# DLVM: A MODERN COMPILER INFRASTRUCTURE FOR DEEP LEARNING SYSTEMS

**Richard Wei**
Departments of Computer Science & Linguistics
University of Illinois at Urbana-Champaign
Urbana, IL 61801
xwei12@illinois.edu

**Lane Schwartz**
Department of Linguistics
University of Illinois at Urbana-Champaign
Urbana, IL 61801
lanes@illinois.edu

**Vikram Adve**
Department of Computer Science
University of Illinois at Urbana-Champaign
Urbana, IL 61801
vadve@illinois.edu

## ABSTRACT

Deep learning software demands reliability and performance. However, many of the existing deep learning frameworks are software libraries that act as an unsafe DSL in Python and a computation graph interpreter. We present DLVM, a design and implementation of a compiler infrastructure with a linear algebra intermediate representation, algorithmic differentiation by adjoint code generation, domain-specific optimizations and a code generator targeting GPU via LLVM. Designed as a modern compiler infrastructure inspired by LLVM, DLVM is more modular and more generic than existing deep learning compiler frameworks, and supports tensor DSLs with high expressivity. With our prototypical staged DSL embedded in Swift, we argue that the DLVM system enables a form of modular, safe and performant frameworks for deep learning.

## 1 INTRODUCTION

Within the deep learning community, most current approaches to neural networks make use of high-level frameworks with a tensor domain-specific language (DSL) such as Torch (Collobert et al., 2011), TensorFlow (Abadi et al., 2016), PyTorch (PyTorch Development Team, 2016), and MXNet (Chen et al., 2015). Traditionally, developers would build a computation graph (or dynamically generate graph nodes) using a DSL and let the framework interpret the computation graph on parallel architectures such as NVIDIA GPUs. While using hand-tuned GPU subroutines usually yields the best performance for complex operators, advanced compiler techniques can be applied to simplify computation, merge high-level operators based on shaping conditions, and fuse compatible element-wise operators to a single kernel to minimize the latency between kernel launches. Recent projects, the TensorFlow XLA compiler (Leary & Wang, 2017) and the NNVM compiler (NNVM, 2017) including TVM (Chen et al., 2017), have begun to apply compiler techniques to deep learning systems, targeting LLVM (Lattner & Adve, 2004) and various back-ends to achieve good performance. However, their design and implementation have not entirely followed established best practices in widely-used compiler frameworks in the industry.

Moreover, some frameworks use operator-overloading algorithmic differentiation (AD) to compute gradients, leaving the gradient computation unoptimizable. The other approach to AD, source code transformation, can produce more efficient code. While frameworks such as TensorFlow already perform AD as a graph transformation and apply various optimizations, their AD transformation is not designed as a transformation pass in the pipeline of their compiler framework, but as part of the DSL library. Making AD part of the compiler framework would greatly simplify the development of DSLs, achieving separation of concerns.

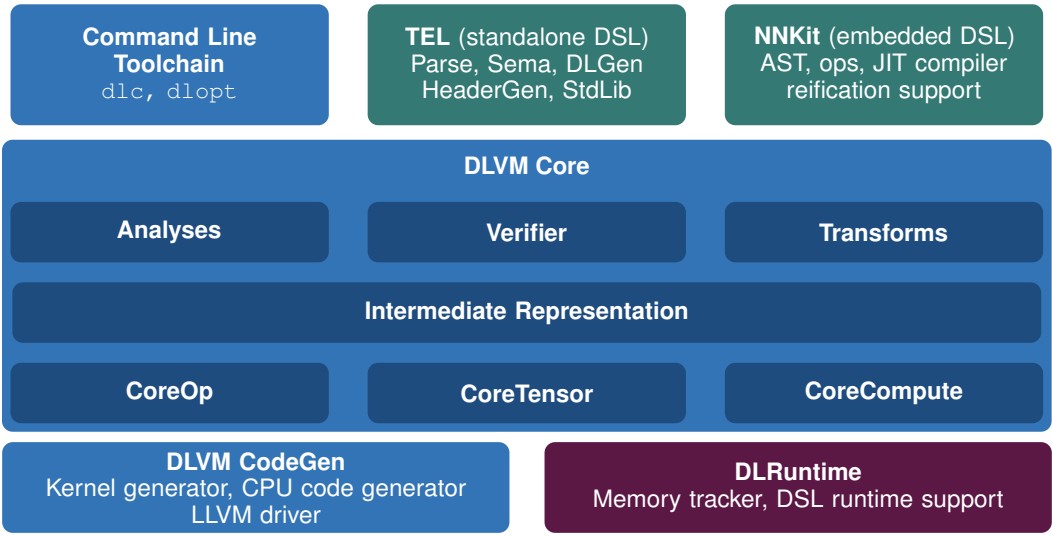

Figure 1: Software stack of the DLVM infrastructure. Blue components are the compiler framework.

We introduce DLVM, a new compiler infrastructure for deep learning systems that addresses shortcomings of existing deep learning frameworks. Our solution includes (1) a domain-specific intermediate representation specifically designed for tensor computation, (2) principled use of modern compiler optimization techniques to substantially simplify neural network computation, including algebra simplification, AD checkpointing, compute kernel fusion, and various traditional compiler optimizations, (3) code generation through a mature compiler infrastructure that allows for transparent targeting of various hardware, and (4) an embedded DSL that supports static analysis, type safety, and natural expression of tensor computation, and has a just-in-time (JIT) compiler targeting DLVM for AD, optimizations, and code generation.

## 2 RELATED WORK

Numerous existing projects provide specialized systems for machine learning but are not closely related to our work. These include Apache SystemML (Ghoting et al., 2011), a high-level language and framework for writing and executing machine learning problems targeting Apache Spark, and TACO (Kjolstad et al., 2017), a C++ library for compiling and optimizing kernels that is more similar to Halide (Ragan-Kelley et al., 2013) than to our work. Our work treats the creation of neural networks as a compilers problem to be addressed using mature compiler techniques. SystemML does not consider this issue at all; TACO does use compiler optimization, but only at a very low level to generate individual kernels.

Two projects closely related to this work are the TensorFlow XLA compiler and the NNVM compiler. The code representation in these frameworks is a "sea of nodes" representation, embedding control flow nodes and composite nodes in a data flow graph. To apply algorithmic differentiation on this IR requires non-standard processing. In contrast, our approach is designed from the start around the idea that a neural network (and its associated tensor computations) is itself a program, which is best optimized through robust application of mature techniques in a principled compilation pipeline. We represent tensor computation in static single assignment (SSA) form with control flow graph, and perform algorithmic differentiation, domain-specific optimizations, general-purpose optimizations, low-level optimizations, and code generation.

XLA takes a similar approach to ours, transforming TensorFlow sub-graphs to XLA's HLO graph and performing optimizations. Our intermediate representation is much more expressive than XLA's by including modular IR components and general-purpose instructions; this enables our approach to support full-fledged DSLs including standalone compiled DSLs and perform more extensive optimizations such as inlining and interprocedual optimizations. Our approach also differs from XLA by representing composite functions such as `min` and `max` directly through primitive instructions

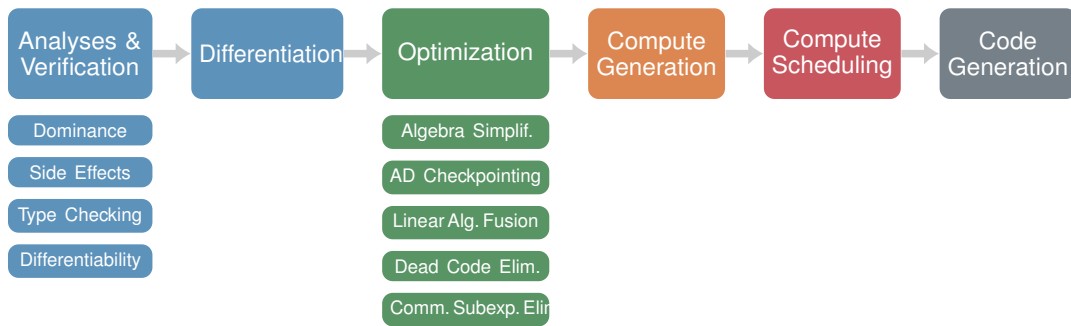

Figure 2: Compilation stages in the DLVM compilation pipeline.

such as `compare` and `select`, which enables us to apply generic AD, and by using SSA form with control flow graph, which allows for reusing battle-tested SSA optimization algorithms in the LLVM community. Importantly, our entire infrastructure was designed from the start around a robust compile-time framework for tensor DSLs, whereas XLA has been adapted around the existing TensorFlow infrastructure with a particular focus on hardware support for Google's Tensor Processing Units (Jouppi et al., 2017).

Where TVM and NNVM are built as a DSL and a graph library in Python with a C++ implementation, DLVM's architecture is closer to LLVM and the Swift Intermediate Language (Groff & Lattner, 2015), having an IR file format and a full-fledged command line toolchain. More specifically, our work differs from NNVM and XLA in the design and presence of an IR that has a textual parsable format, a module/function/basic block hierarchy, custom type declarations and memory semantics. The textual IR enables robust unit testing via FileCheck, which is used extensively in LLVM and most LLVM-based compilers. Moreover, DLVM and its associated DSLs are implemented entirely in Swift, a safe systems programming language, and thus have an elegantly compact codebase and type-safe APIs.

## 3  DLVM

Deep Learning Virtual Machine (DLVM) is a compiler infrastructure designed for modern deep learning systems.[1] DLVM is designed to apply a multi-stage compiler optimization strategy to both high-level linear algebra and low-level parallelism, perform domain-specific transformations, relieve the overhead in front-end languages, and serve as the host for research and development of DSLs for deep learning. The complete DLVM software stack, including sample front-end deep learning DSLs, is shown in Figure 1 on the preceding page.

Figure 2 illustrates the major stages in the DLVM compilation pipeline. The DLVM compilation stages address algorithmic differentiation, domain-specific optimizations, general-purpose optimizations, and static code generation targeting a variety of compute architectures. The *raw* and *optimizable* stages allow constructs for high-level tensor operations and various high-level optimizations. The *compute* and *schedule* stages allow constructs for low-level array operations lowered from tensor operations in high-level stages, borrowing the design from Halide (Ragan-Kelley et al., 2013).

The DLVM Intermediate Representation (IR) is the core language of the system. It uses static single assignment (SSA) form, control flow graphs, high-level types including a first-class tensor type, and a set of linear algebra operators combined with a general-purpose instruction set (see Table 1). The system enables a wide variety of domain-specific analyses and transformations, such as reverse-mode AD, AD checkpointing, algebra simplification and linear algebra fusion.

To demonstrate how DLVM helps the development of domain-specific languages (DSLs), in Section 3.4 we present one prototypical staged DSL: NNKit. NNKit features safe and natural expression of tensor computation alongside the host program, and targets DLVM for differentiation, optimizations and static code generation.

---

[1]Code for DLVM is available at `https://github.com/dlvm-team`

| Kind | Example |
|------|---------|
| Element-wise unary | `tanh %a: <10 x f32>` |
| Element-wise binary | `power %a: <10 x f32>, %b: 2: f32` |
| Dot | `dot %a: <10 x 20 x f32>, %b: <20 x 2 x f32>` |
| Reduce | `reduce %a: <10 x 30 x f32> by add along 1` |
| Transpose | `transpose %m: <2 x 3 x 4 x 5 x f32>` |
| Slice | `slice %a: <10 x 20 x i32> from 1 upto 5` |
| Compare | `gt %a: <10 x 20 x bool>, %b: <1 x 20 x bool>` |
| Data type cast | `dataTypeCast %x: <10 x f32> to f64` |
| Function application | `apply %foo(%x: f32, %y: f32): (f32, f32) -> f32` |
| Branch | `branch 'block_name(%a: i32, %b: i32)` |
| Conditional branch | `conditional %cond: bool then 'bb0() else 'bb1()` |
| Shape cast | `shapeCast %a: <1 x 40 x f32> to 2 x 20` |

Table 1: This table illustrates a selection of the instructions in the DLVM virtual instruction set. The instruction set includes linear algebra operations such as `tanh` and `dot` in addition to control flow instructions such as `branch`.

## 3.1 DLVM CORE

DLVM Core contains essential components for an optimizing compiler: IR, pass manager, and passes (see Figure 1 on page 2). The DLVM IR consists of a virtual instruction set, control flow graph and data flow representation. Passes are functions that traverse the intermediate representation of a program, either producing useful results as analyses of the program (analysis passes), or mutating the program for differentiation and optimizations (transform passes).

### 3.1.1 A DOMAIN-SPECIFIC COMPILER INTERMEDIATE REPRESENTATION FOR DLVM

Inspired by the LLVM IR (Lattner & Adve, 2004) and the Swift Intermediate Language (Groff & Lattner, 2015), DLVM IR is a graph-based, modular code representation, with both an in-memory format and a textual format. The code representation has a hierarchy of abstractions: **module**, **function**, **basic block**, and **instruction**. An instruction is the minimal unit of code that operates on values, which can be globals, function arguments or temporary virtual registers produced by instructions. Each module contains a collection of type definitions, global values and functions. Each function has a control flow graph formed by basic blocks and control flow edges. Each basic block contains an ordered list of instructions with data dependencies forming a directed acyclic graph.

The DLVM IR has a high-level type system with tensor as a first-class type. The DLVM virtual instruction set includes domain-specific primitive math operators, as well as general-purpose instructions for memory management, control flow and function application. Domain-specific instructions include element-wise unary operators, such as `tanh` and `negate`, element-wise binary operators, such as `add` and `power`, and complex operators such as `dot`, `transpose`, and `convolve`. All element-wise binary operators support broadcasting. A sample of DLVM IR code is shown in Figure 3 on the next page.

The DLVM instruction set does not include composite math functions such as `softmax`, `sigmoid`, `min` or `max`. All of these functions can be composed of primitive math instructions and control flow constructs. This design allows for the standard AD algorithm to be applied to any differentiable program, with no need for special handling of composite cases.

### 3.1.2 DOMAIN-SPECIFIC COMPILER PASSES FOR DLVM

DLVM has a full-fledged pass infrastructure, performing various analyses and two kinds of transformations: differentiation and optimization. Differentiation constructs function definitions from gradient declarations using adjoint code generation (see Section 3.1.3 below). Optimization is then performed on the resulting IR, maximizing the code performance. Optimizations include domain-specific optimizations, such as algebra simplification, linear algebra fusion, matrix multiplication reordering, and AD checkpointing, and traditional compiler optimizations.

```
module "my_module"
stage raw

// Representing function foo(x, w, b) = dot(x, w) + b
func @foo: (<1 x 784 x f32>, <784 x 10 x f32>, <1 x 10 x f32>)
        -> <1 x 10 x f32> {
'entry(%x: <1 x 784 x f32>, %w: <784 x 10 x f32>, %b: <1 x 10 x f32>):
    %v0 = dot %x: <1 x 784 x f32>, %w: <784 x 10 x f32>
    %v1 = add %v0: <1 x 10 x f32>, %b: <1 x 10 x f32>
    return %v1: <1 x 10 x f32>
}

// Gradient of @foo with respect to all arguments
[gradient @foo]
func @foo_grad: (<1 x 784 x f32>, <784 x 10 x f32>, <1 x 10 x f32>)
        -> (<1 x 784 x f32>, <784 x 10 x f32>, <1 x 10 x f32>)

// Gradient of @foo with respect to arguments 1 and 2
// Keeping original output 0
// Seedable, able to take back-propagated gradient as a seed for AD
[gradient @foo wrt 1, 2 keeping 0 seedable]
func @foo_grad_3:
    (<1 x 784 x f32>, <784 x 10 x f32>, <1 x 10 x f32>, <1 x 10 x f32>)
    -> (<784 x 10 x f32>, <1 x 10 x f32>, <1 x 10 x f32>)
```

Figure 3: Example code in DLVM intermediate representation. Note that some functions are annotated as defining the `gradient` of another function with respect some or all arguments. The body of these gradient functions will be automatically generated.

Since DLVM IR is aware of mathematical operators such as `tanh` and `power`, the algebra simplification pass can find and simplify certain mathematical operations that are expensive or redundant. For example, $x^2$ can be simplified to $x \odot x$ ($\odot$ stands for element-wise multiplication), and $x^0$ can be simplified to constant 1. Matrix multiplication reordering is another classic optimization that minimizes the number of sub-operations in a chain of matrix multiplications with different dimensionality, based on matrix multiplication's associativity.

Since the DLVM optimizer is aware of linear algebra operations with static dimensionality, maximizing the performance by fusing verbose linear operations into a single matrix multiplication is beneficial as well. For example, it is very common to encounter expressions of the form $\mathbf{Wx} + \mathbf{b}$. When unoptimized, the matrix multiplication and the addition will be parallelized separately. Since launching compute kernels separately can be expensive, DLVM performs linear algebra fusion, which transforms subexpressions involving both matrix multiplication and element-wise operations into a single matrix multiplication instruction on padded tensors. Besides the simple pattern like an addition of matrix multiplication and a vector, we can apply the same approach to a polynomial of multiple matrix multiplications, turning the polynomial into a single matrix multiplication. For example, in a simple recurrent neural network (RNN), each cell of the recurrence is a feed forward neural network that takes two inputs: $\mathbf{x}_t$, the input local to the current timestep, and $\mathbf{h}_t$, the hidden state carried along the recurrence. The linear algebra fusion pass can simplify operations in $\mathbf{h}_t = f(\mathbf{Wx}_{t-1} + \mathbf{Uh}_{t-1} + \mathbf{b})$ from two matrix multiplications and two additions into a single matrix multiplication. A more aggressive, interprocedural version of linear algebra fusion can optimize parameter passing and memory allocation, so that the entire concatenated matrix can be created and passed around in the first place without reallocation.

### 3.1.3 ALGORITHMIC DIFFERENTIATION THROUGH ADJOINT CODE GENERATION

Algorithmic differentiation (AD), also known as automatic differentiation, encompasses a family of a well-known techniques for algorithmically obtaining the derivatives of a function $f : \mathbf{x} \in \mathbb{R}^n \to \mathbf{y} \in \mathbb{R}^m$ (Naumann, 2011). The function $f$ can be represented as a directed acyclic computation graph representing the composition of elementary computational operations for which the respective

derivatives are well known. The partial derivative $\frac{\partial y_j}{\partial x_i}$ can be computed through recursive applications of the chain rule, either in the forward direction (corresponding to a bottom-up traversal of the computation graph) or in the backward direction (corresponding to a top-down traversal of the computation graph). The former approach, called forward-mode or tangent-mode AD, is used in some research communities, most commonly when $m \gg n$ (Goodfellow et al., 2016). The latter approach, which includes the back-propagation algorithm (Rumelhart et al., 1986) as a special case, is called reverse-mode or adjoint-mode AD, and encompasses the techniques most commonly used for training the weights in neural networks.

In DLVM, the differentiation pass is responsible for performing reverse-mode AD. This pass is responsible for generating DLVM IR code that calculates the derivative of a differentiable function. A function is marked as being automatically differentiable via **gradient declarations**. A gradient declaration is a function in a module that is declared with its mathematical relation with another function in the module and no function body. The function `@foo_grad` in Figure 3 is an example of such a function. Gradient declarations are configurable, e.g. specifying arguments to differentiate with respect to, keeping original outputs, and toggling seedability to accept back-propagated gradients. The differentiation pass, when applied, canonicalizes every gradient declaration in the module to a normal function definition with basic blocks and instructions. The canonicalization process first copies basic blocks and instructions from the original function to the new function body, and then applies adjoint code generation to the function. Unlike many of the existing deep learning frameworks, AD in DLVM is a source code transformation, not interpretation (operator overloading) over the same program. This makes the compiler able to perform optimizations on the gradient computation separately and enables higher order differentiation.

Given a differentiable function $f(\mathbf{x}_1, \mathbf{x}_2, \ldots, \mathbf{x}_n)$, this pass creates a new function that computes the Jacobian $\mathbf{J}_f$. This approach to AD has several advantages with respect to AD performed by operator overloading / graph interpretation. Unlike operator overloading, the gradient function produced by AD is a standalone function that sits uniformly alongside other functions in an IR module, representationally unrelated to the original function. The generated function takes original inputs and produces a tuple of partial derivatives with respect to the inputs. In AD, not all values in the forward evaluation will necessarily be used to compute derivatives. In DLVM, unused operations can be easily eliminated by the aggressive dead code elimination pass in the compilation pipeline (see Section 3.1.4). In addition, an AD-specific optimization technique called checkpointing can further reduce memory consumption during gradient computation.

AD in DLVM is configurable. The front-end can choose to differentiate a function with respect to selected arguments, to keep some of the outputs of the original function, to apply differentiation to a specific output when there are multiple return values, or to enable the function to accept back-propagated gradients (seeds) through function composition, all by gradient declarations. If the function returns multiple values in a tuple, the gradient declaration can also specify which tuple element to differentiate. Our approach to AD is implemented as a transformation from one function to another function. This approach also makes higher-order differentiation possible; this can be accomplished by declaring a higher-order gradient function that differentiates the original gradient function.

### 3.1.4 General-purpose Optimizations for DLVM

General-purpose optimizations refer to traditional compiler optimizations applied to DLVM IR. These optimizations are important at the DLVM stage in the compilation pipeline, since linear algebra computation can be highly optimized or eliminated before they get lowered to LLVM IR which contain parallel execution and low-level information that prevent LLVM optimizations from identifying high-level patterns. Some of these optimizations are aggressive dead code elimination, common subexpression elimination, and sparse conditional constant propagation. Applying such optimizations on gradient computation is not feasible in other approaches to AD that use graph interpretation (operator overloading), because the forward pass and the backward pass are tied together in a single graph; mutating either evaluation pass will alter the semantics of the other.

### 3.2 Code generation

Two major design goals of DLVM are the ability to target multiple heterogenous parallel architectures from the same front-end DSL code (and corresponding DLVM IR), and the ability to perform aggressive optimizations on lowered programs. In order to attain these goals, DLVM code generation transforms DLVM IR into LLVM IR. LLVM is a robust and mature compiler infrastructure with multiple back-ends, including NVIDIA GPUs. Many high-level DLVM IR linear algebra instructions over tensors abstract lower-level operations. The DLVM compiler transforms the high-level DLVM IR into lower-level stages and ultimately into calls to BLAS and compute kernels in LLVM IR. Existing LLVM utilities are used to compile the generated LLVM IR to the final binary.

In order to take full advantage of a variety of emerging heterogeneous parallel architectures, we plan for future versions of DLVM to target the IR of HPVM (Srivastava et al., 2016), a higher-level heterogeneous compiler extension to LLVM IR that allows for transparent targeting of diverse architectures from a data flow graph.

### 3.3 DLVM command line toolchain

The front-end software associated with each DSL (see Section 3.4) is responsible for generating a DLVM IR for a given source language program to be compiled. The DLVM compiler infrastructure itself is a compiler from DLVM IR to LLVM IR, therefore having a command line toolchain is necessary for verifying, transforming and compiling batches of DLVM IR files (`*.dl`). Unlike XLA and NNVM/TVM, which only provide a Python/C++ interface to their users, DLVM provides a command line interface like any industry-standard compiler.

The DLVM optimizer utility, `dlopt`, accepts `*.dl` IR files and applies user-specified optimization passes on them. The DLVM compiler driver, `dlc`, accepts `*.dl` IR files and performs user-specified tasks, such as verification, differentiation, optimization passes, stage lowering passes, and code generation; the driver invokes the DLVM core library to achieve these tasks. Because of having a textual format of the IR, the DLVM framework can easily make use of the LLVM Integrated Tester (lit) and FileCheck to perform robust unit testing. In future development, we plan to introduce a DLVM bitcode format for compact storage and high-throughput processing of DLVM code.

### 3.4 Neural network DSLs

Most of existing deep learning DSLs are embedded in a dynamically typed scripting language such as Python and Lua. While these scripting languages provide flexibility and a large number of libraries for scientific computing, they can act as a barrier between lightweight prototyping code and systematic production code. This barrier significantly reduces the reliability of ML software, resulting in suboptimal programming experience and unnecessarily effortful development cycles.

In software engineering, a proven approach to tackle this problem is language and compiler technologies, starting from a language that is amenable to static analysis. A well-designed deep learning DSL should support the needs of deep learning software developers by providing a safe environment for rapid prototyping, while simultaneously generating highly efficient code for training and inference. DSLs in a scripting language can easily achieve rapid prototyping, but they are generally incapable of providing a safe environment with optimal performance. We believe that the best solution is DSLs embedded in a type-safe, type-inferring programming language that is both fast and easy to learn. In our initial release of DLVM, we provide one such DSL, both as a proof-of-concept and as a reference implementation that showcases the capabilities of DLVM as a platform for deep learning DSL development.

NNKit is a staged DSL embedded in Swift, featuring natural expression of tensor computation alongside the host program without losing static analyses and optimizations on the tensor program. Inspired by Lightweight Modular Staging (Rompf & Odersky, 2010), NNKit leverages the static type system of Swift to guarantee type safety of the DSL. Tensor types are wrapped in Rep<T>, meaning the representation of some computation that produces data of type T. Tensor operators overloaded for Rep are essentially AST builders for delayed evaluation. Instead of generating computation nodes at runtime and performing operator-overloading AD like PyTorch or TensorFlow Eager (Google Brain Team, 2017), NNKit tensor computations are staged once during the lifetime of the host program. At invocation time of staged functions, NNKit emits shape-specialized DLVM IR and leverages DLVM

```
// Staged function representing f(x, w, b) = dot(x, w) + b
let f: Rep<(Float2D, Float2D, Float2D) -> Float2D> =
    lambda { x, w, b in
            x • w + b
    }

// Staged function 'g', type-inferred from 'f'
let g = lambda { x, w, b in
    let linear = f[x, w, b] // staged function application
    return tanh(linear)
}

// Gradient of 'g' with respect to arguments 'w' and 'b'
let dg = gradient(of: g, withRespectTo: (1, 2), keeping: 0)
// 'dg' has type:
// Rep<(Float2D, Float2D, Float2D) -> (Float2D, Float2D, Float2D)>

// Call staged function on input data 'x', 'w' and 'b'
let (dg_dw, dg_db, result) = dg[x, w, b]
// At runtime, 'dg' gets just-in-time compiled though DLVM,
// and computes ( dg/dw, dg/db, g(x, w, b) )

// Second order derivative of 'g' with respect to 'w'
let d2g_dw2 = gradient(of: dg, from: 0, withRespectTo: (1))
// 'd2g_dw2' has type:
// Rep<(Float2D, Float2D, Float2D) -> Float2D>
```

Figure 4: Example code in Swift using NNKit, a staged DSL targeting DLVM.

to perform AD, optimizations, and low-level code generation. A sample of Swift code using NNKit is shown in Figure 4.

The NNKit just-in-time compiler has four important phases: The first phase, expression staging, produces an unshaped graph IR of the tensor computation. The second phase, shape specialization, prepares to generate statically shaped DLVM IR for staged functions when they are applied to shaped tensors. The third phase, lowering, generates DLVM IR and passes it through DLVM, producing a dynamic library containing a function symbol. The final phase, function reification, loads the binary and wraps the low-level function to a Swift function.

We anticipate other existing deep learning frameworks, such as TensorFlow, could be adapted to use DLVM as a back-end to their tensor math DSLs.

## 4 CONCLUSION

The deep learning research community has a rich variety of available frameworks. While two existing projects have attempted a compilers approach to deep learning frameworks, and have respectively achieved good integration with existing systems (TensorFlow XLA) and good performance (NNVM + TVM), their design philosophies have not entirely followed established best practices in optimizing compiler design. While well intentioned, the remaining vast majority of other frameworks have failed to observe that the problem of front-end DSLs, algorithmic differentiation, and converting a neural network into efficient executable code is, at its core, a compilers problem. As a result, important issues of extensibility and optimization have been addressed in less than optimal fashion in such frameworks. Nevertheless, several such frameworks have achieved wide adoption. We believe that the principled application of optimizing compiler techniques will lead to substantial improvements in the tools available to deep learning researchers. DLVM and its associated front-end DSLs have a major role to play in this future. Our existing implementation supports reverse-mode AD in the core language, and utilizes LLVM to target NVIDIA GPUs. In our ongoing work, we plan to substantially increase the number of supported hardware architectures by utilizing HPVM as an additional back-end, and explore more advanced AD techniques such as mixing forward and reverse modes.

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
