# OpenReview forum: "DLVM: A modern compiler infrastructure for deep learning systems"
_ICLR.cc/2018/Conference — Invite to Workshop Track_

### Official Review · AnonReviewer3 · 2017-11-22
**Modern-day neural networks require code generation (to generate the backward pass) and a host of other optimizations to be run effectively. The paper proposes to do this using best practices in the compilers and programming languages communities as opposed to the ad hoc manner adopted by current deep learning engines.**

**Rating:** 5
**Confidence:** 4

**Review:**

Modern-day deep learning engines (e.g., tensorflow, caffe) perform code generation (to generated the backward pass) and a host of other optimizations to run today's deep learning workloads. Unfortunately, the manner in which they go about doing this is ad hoc and does not adopt best practices developed in the compilers and programming languages communities. The obvious consequence is missed opportunities for optimizing deep learning workloads. This paper proposes to fix this by re-designing from ground up the deep learning engine placing particular focus on code generation, compilation (e.g., type checking), optimization (e.g., fusion, matrix chain reordering, common subexpression elimination) etc. Unfortunately, the paper falls short in two significant respects: It does not adequately cite related work and it does not present any experiments to quantify the benefits they claim will be achieved by their new compiler.

Pros:
- The paper proposes a very relevant and timely proposal to design a modern day deep learning compiler framework.
- Their design includes a number of optimizations that are missing from currently available deep learning engines which can lead to significant benefits.

Cons:
- Related work is not adequately referenced. Here are two (others should be easy to find): Apache SystemML (https://systemml.apache.org/), TACO (http://tensor-compiler.org/)
- Experiments section is conspicuous by its absence. They provide no micro benchmarks or end-to-end deep learning use cases to quantify the benefits of their compiler vs. some of the currently available ones.

---

### Official Review · AnonReviewer2 · 2017-11-24
**The paper presents a compiler framework for domain-specific languages (DSLs) targeting deep learning systems. The paper describes the overall structure, the different compilation stages, and also provides a small example. This type of frameworks is very useful and can have a significant impact in the community. However, no evaluation of the proposed framework is done in the paper.**

**Rating:** 7
**Confidence:** 4

**Review:**

Deep learning is a technique that has attracted a lot of attention. Typically, when using a deep learning framework we describe the network using some kind of computation graph, e.g., as in TensorFlow. A drawback is that the execution performance can be limited, e.g., due to run-time interpretation of the computation graph.

This paper takes a different approach and presents a compiler framework that allows definition of domain-specific languages (DSLs) for deep learning system, defines a number of compilation stages that can take advantage of standard compiler optimizations as well as specialized optimizations for neural networks using an intermediate representation, and also a back-end. Thus a computation graph is compiled directly to binary code, which increases the performance. For example, the compiler infrastructure enables optimization over multiple kernels using kernel fusion.

I find the paper very interesting and the findings can have a significant impact on how we develop deep learning systems in the future. The paper addresses an important problem, is very well written, and is easy to follow. The different optimization stages are well describe and also motive why they improve performance over existing techniques. The intention is to provide the framework as open source in the future.

The main drawback of the paper is the lack of evaluation. Although the framework is well described, its application and use are only demonstrated with a very small code example. No comparison with existing frameworks is done, and no evaluation of the actual performance is done.

---

### Official Review · AnonReviewer5 · 2017-12-29
**A tensor compiler (as opposed to a DSL inside a general purpose language)**

**Rating:** 5
**Confidence:** 3

**Review:**

The success of Deep Learning is, in no small part, due the development of libraries and frameworks which have made building novel models much easier, faster and less error prone and also make taking advantage of modern hardware (such as GPUs) more accessible. This is still a vital area of work, as new types of models and hardware are developed.

This work argues that prior solutions do not take advantage of the fact that a tensor compiler is, essentially, just a compiler. They introduce  DLVM (and NNKit) which comprises LLVM based compiler infrastructure and a DSL allowing the use of Swift to describe a typed tensor graph. Unusually, compared to most frameworks, gradients are calculated using source code transformation, which is argued to allow for easier optimization.

This paper is not well-adapted for an ICLR audience, many of which are not experts in compilers or LLVM. For example, the Figure 3, table 1 would be benefit from being shorter with more exposition on what the reader should understand and take away from them.

The primary weakness of this work is the lack of careful comparison with existing framework. The authors mention several philosophical arguments in favor of their approach, but is there a concrete example of an model which is cumbersome to write in an existing framework but easy here? (e.g. recent libraries pytorch, TF eager can express conditional logic much more simply than previous approaches, its easy to communicate why you might use them). Because of this work seems likely to be of limited interest to the ICLR audience, most of which are potentially interested users rather than compiler experts. There is also no benchmarking, which is at odds with the claims the compiler approaches allows easier optimization.

One aspect that seemed under-addressed and which often a crucial aspect of a good framework, is how general purpose code e.g. for loading data or logging interacts with the accelerated tensor code.

---

> ### Comment · AnonReviewer5 · 2018-01-16
> **No change in ratings**
>
> I appreciate the author's response (I'm a little confused why everything is promised in a future update, e.g. clarifications and improvements in the paper, is there no opportunity to update the pdf on openreview? This would make it easier to appreciate these changes).
>
> However, I still feel the lack of comparison with other frameworks makes this work feel unfinished, or a position paper that will be of limited interested to an ICLR audience. The author's argue that even without empirical evaluation it will still be of interest, but I think it will be of more specialized interest to a small subset who are writing frameworks. As a researcher, I don't come away with any feeling for 'in this situation you should really consider using this tool.'
>
> For this reason I'm leaving my evaluation unchanged.

---

### Author Response · Authors · 2018-01-12
**Response to reviews**

First, thank you to the reviewers for their time and effort in reviewing this submission. We very much appreciate their attention and their efforts.

AnonReviewer5 wrote: "This paper is not well-adapted for an ICLR audience, many of which are not experts in compilers or LLVM. For example, the Figure 3, table 1 would be benefit from being shorter with more exposition on what the reader should understand and take away from them."

We agree that most ICLR attendees are not experts in compilers or LLVM. It is for that very reason that we believe this paper would be a valuable addition to ICLR. Many in the ICLR audience make heavy use of deep learning toolkits, despite any shortcomings of those toolkits. We argue that our paper is well situated as a position paper designed to make ICLR attendees more aware of the inherent design shortcomings of existing deep learning toolkits, and present a sound alternative. If accepted, we would abbreviate Figure 3 and Table 1, and provide additional textual exposition on what the reader should take away.


AnonReviewer3 wrote: "Related work is not adequately referenced. Here are two (others should be easy to find): Apache SystemML (https://systemml.apache.org/), TACO (http://tensor-compiler.org/)"

Apache SystemML is a high-level language and framework for writing and executing machine learning problems, especially targeting Apache Spark. TACO is much more similar to Halide than it is to our work. TACO is a C++ library for compiling and optimizing kernels. Neither TACO nor SystemML are closely related to our work. Our work argues that deep learning (and in particular the creation of neural network topologies) is itself a compilers problem, and should be addressed using mature compiler techniques. SystemML does not consider this issue at all. TACO does use compiler optimization, but only at a very low level to generate individual kernels. There is existing work that is related to ours, namely XLA, TVM, and NNVM. In Section 2, we examine those systems and describe in detail how our work differs from those.


AnonReviewer2 wrote: "I find the paper very interesting and the findings can have a significant impact on how we develop deep learning systems in the future. The paper addresses an important problem, is very well written, and is easy to follow. ... The main drawback of the paper is the lack of evaluation. Although the framework is well described, its application and use are only demonstrated with a very small code example."

We concur with this reviewer that our approach is likely to have a substantial impact on the development of deep learning systems. Given this likely impact, we believe that there is significant value in presenting this paper to the ICLR community, despite the fact that we were not yet able to present quantifiable evaluation results.

---

> ### Comment · AnonReviewer3 · 2018-01-15
> **Given that most ICLR folks don't have a compiler background, may still help if the blurb about SysML/TACO is included in the paper.**
>
> The title of my comment says it all. Given that most ICLR audience do not have a background to make nuanced distinctions among the various deep learning engines available (myself included), do the authors feel it would help to include the blurb about SystemML and TACO into the draft/paper?

---

> > ### Author Response · Authors · 2018-01-15
> > **Yes, we would be happy to add additional related work descriptions**
> >
> > If the paper is accepted, we will add a description based on the above blurb to the related work section, describing how our work differs from SysML and TACO.

---

### Decision · Program_Chairs · 2018-01-29
**ICLR 2018 Conference Acceptance Decision**

**Decision:**

Invite to Workshop Track

**Comment:**

This is a fascinating paper, and representative of the sort of work which is welcome in our field and in our community. It presents a compiler framework for the development of DSLs (and models) for Deep Learning and related methods. Overall, reviewers were supportive of and excited by this line of work, but questioned its suitability for the main conference. In particular, the lack of experimental demonstrations of the system, and the disconnect between domain-specific technical knowledge required to appreciate this work and that of the average ICLR attendee were some of the main causes for concern. It is clear to me that this paper is not suitable for the main conference, not due to its quality, but due to its subject matter. I would be happy, however, to tentatively recommend it for acceptance to the workshop as this topic deserves discussion at the conference, and this would provide the basis for a useful bridge between the compilers community and the deep learning community.